# Joint Cartilage in Long-Duration Spaceflight

**DOI:** 10.3390/biomedicines10061356

**Published:** 2022-06-08

**Authors:** Bergita Ganse, Magali Cucchiarini, Henning Madry

**Affiliations:** 1Werner Siemens Foundation Endowed Chair of Innovative Implant Development (Fracture Healing), Clinics and Institutes of Surgery, Saarland University, 66421 Homburg, Germany; 2Department of Trauma, Hand and Reconstructive Surgery, Clinics and Institutes of Surgery, Saarland University Medical Center, 66421 Homburg, Germany; 3Center of Experimental Orthopaedics, Saarland University Medical Center, 66421 Homburg, Germany; magali.madry@uks.eu (M.C.); henning.madry@uks.eu (H.M.)

**Keywords:** astronaut, cosmonaut, taikonaut, immobilization, unloading, weightlessness, microgravity, musculoskeletal system, osteoarthritis, bed rest

## Abstract

This review summarizes the current literature available on joint cartilage alterations in long-duration spaceflight. Evidence from spaceflight participants is currently limited to serum biomarker data in only a few astronauts. Findings from analogue model research, such as bed rest studies, as well as data from animal and cell research in real microgravity indicate that unloading and radiation exposure are associated with joint degeneration in terms of cartilage thinning and changes in cartilage composition. It is currently unknown how much the individual cartilage regions in the different joints of the human body will be affected on long-term missions beyond the Low Earth Orbit. Given the fact that, apart from total joint replacement or joint resurfacing, currently no treatment exists for late-stage osteoarthritis, countermeasures might be needed to avoid cartilage damage during long-duration missions. To plan countermeasures, it is important to know if and how joint cartilage and the adjacent structures, such as the subchondral bone, are affected by long-term unloading, reloading, and radiation. The use of countermeasures that put either load and shear, or other stimuli on the joints, shields them from radiation or helps by supporting cartilage physiology, or by removing oxidative stress possibly help to avoid OA in later life following long-duration space missions. There is a high demand for research on the efficacy of such countermeasures to judge their suitability for their implementation in long-duration missions.

## 1. Introduction

Long-duration spaceflight, such as flights to Mars and beyond, is a major challenge for the human body and especially the musculoskeletal system [1,2]. Challenges for the musculoskeletal system include not only the adaptation to immobilization and unloading in microgravity, but also damage from radiation exposure [3,4]. In addition, adaptations to inactivity occur due to psychological and behavioral changes, and alterations of the central nervous system related to isolation and confinement [5,6]. Further challenges include changes in immunological reactions and the immune status [7,8] and differences in locomotion when walking on a planet or moon with lower gravity than on Earth [9,10]. The musculoskeletal system is sensitive to changes in the biomechanical environment, and prolonged unloading during exposure to microgravity leads to disuse osteoporosis [11,12], losses in skeletal muscle mass, peak force and power [13,14], and changes in intervertebral disc and articular cartilage physiology [15,16]. Following return to Earth, most musculoskeletal tissues seem to recover well; however, some long-term studies have identified incomplete recovery of trabecular bone mineral density (BMD) and architecture [17,18]. Clinical musculoskeletal problems during spaceflight include Space Adaptation Back Pain (SABP) caused by spinal elongation due to intervertebral disc swelling [19,20], injuries [21], as well as the bends and dysbaric osteonecrosis in the context of decompression sickness that mainly occurred during spacewalks [22,23,24].

Spaceflight-related changes in joint cartilage are currently not among the highest priority human health risks for a mission to Mars, as defined by the National Aeronautics and Space Administration (NASA), as they are no direct threat to mission success [25]. Only very few articles are available that deal with cartilage physiology in space. A Pubmed search for the terms ‘spaceflight joint cartilage’ only revealed 19 hits on 5 June 2022. However, research on articular cartilage and the risk for osteoarthritis (OA) is relevant for the later life of spaceflight participants [26]. Indeed, in rodents, space flight has been shown to induce an arthritic phenotype in the articular cartilage of the knee and degeneration of the meniscus [27]. Given the fact that, apart from total joint replacement or resurfacing, currently no treatment exists for late-stage OA, countermeasures may be needed to avoid cartilage damage during long-duration missions [28]. To plan such countermeasures, it is important to assess if and how joint cartilage and the adjacent structures, such as the subchondral bone, menisci, and ligaments are affected by long-term unloading, reloading, and radiation, and to which degree countermeasures are able to mitigate negative effects. Moreover, all physiological effects related to musculoskeletal unloading and re-loading are of similar clinical relevance for immobile patients, and it is therefore highly desirable to transfer the knowledge gained in the context of spaceflight to clinical medicine. For these reasons, this review summarizes the current literature on joint cartilage alterations in spaceflight, including human, animal, and cell research, discusses countermeasures, identifies knowledge gaps, and points out where further research is needed to give planners the required insights to design and plan interventions for long-term space missions.

## 2. Pathophysiology and Biomarkers of Joint Cartilage Unloading

Articular cartilage is sensitive to mechanical load and shows a high level of mechanoadaptation, with disuse of joints causing profound volume losses in the articular cartilage [29]. Exposure to microgravity leads to a decrease in cartilage thickness that is associated with a loss of glycosaminoglycans (GAGs), highly polar and negatively charged macromolecules [30,31,32,33,34,35]. GAGs play a major role in attracting water into the cartilage [36,37]. Disuse atrophy of articular cartilage accelerates structural progression of OA [38], a condition that is influenced by both inflammatory and biomechanical factors and associated with pain and loss of mobility [26,39]. In rats, during extended knee immobilization in flexion, joint cartilage is progressively replaced by bone tissue [40,41]. In early OA, the articular cartilage surface increasingly shows fibrillation and vertical fissures [42]. Biomechanically, mainly trauma with cartilage injury and increased loads induce OA, such as in obesity or joint malalignment. In addition, genetic factors affect OA susceptibility [43].

A great number of joint cartilage biomarkers exist [44], but only some of them have been studied in the spaceflight context. Relevant inflammatory markers of OA include cytokines (interleukins IL-1, IL-6, IL-10, IL-17, tumor necrosis factors TNF-α and TNF-γ), cylooxygenase-2 (COX2), nitric oxide synthase-2 (NOS2), prostaglandin E2 (PGE2), and nitric oxide (NO). Cartilage oligomeric matrix protein (COMP) and matrix metalloproteinases (in particular MMP-3) are mechano-sensitive cartilage biomarkers, and serum levels were reported to drop in immobilization and bed rest [45]. COMP, also known as thrombospondin-5, is an extracellular matrix protein expressed primarily in cartilage, ligaments, and tendons, and serves as a biomarker of OA [46,47]. COMP serum levels increase after physical exercise [48]. Matrix metalloproteinases are a group of enzymes that play a role in tissue remodeling and repair by degrading the extracellular matrix. MMP-3, MMP-8, and MMP-9 are exclusively expressed in pathological conditions [49,50]. MMP-1 and MMP-8 are produced by synovial cells and chondrocytes in superficial layers of articular cartilage. In addition to the matrix metalloproteinases, the C-terminal cross-linked telopeptide of type II collagen (CTX-II), an index of total type II collagen degradation, and human type IIA collagen *N*-propeptide (PIIANP) are further biomarkers of cartilage matrix degradation [51,52].

Apart from cartilage biomarkers, also markers of the subchondral bone are of interest in OA research, as the subchondral bone is increasingly remodeled during the course of the disease [53]. While CTX-I is a biomarker of subchondral bone resorption, human procollagen I N-terminal propeptide (PINP) indicates subchondral bone formation [53].

In addition to these biomarkers, imaging can be employed to study changes in articular cartilage. Magnetic resonance imaging (MRI) is particularly suitable for cartilage research, as not only cartilage thickness, but also the composition can be studied, such as changes in proteoglycan and collagen composition [54]. Among the MRI techniques, delayed gadolinium-enhanced MRI of cartilage (dGEMRIC) is particularly capable of reflecting the cartilage proteoglycan concentration [55,56]. An inverse correlation was shown for T1ρ relaxation times and proteoglycan concentrations, and there is also a weak correlation between T2 relaxation times and proteoglycans [54]. Calcified cartilage, the subchondral bone plate, and the trabecular bone of the subarticular spongiosa are mineralized tissues that can clinically best be analyzed by computed tomography (CT) or peripheral quantitative computer tomography (pQCT) [57,58].

## 3. Bed Rest and Other Analogue Study Settings

Bed rest in 6° head down tilt (HDT) is a popular spaceflight analogue and serves to study the effects of potential countermeasures on the human body [59]. Healthy participants are usually subjected to 6° HDT bed rest in a highly standardized environment that produces similar cardiovascular and musculoskeletal changes as microgravity (https://www.nasa.gov/sites/default/files/atoms/files/bed_rest_studies_complete.pdf, accessed on 5 June 2022). Participants are not allowed to sit, stand, or exercise during the entire bed rest period. At least one of their shoulders has to touch the mattress at any time. In a 14-day bed rest study, via MRI, a loss in average cartilage thickness of −8% (3.08 to 2.82 mm) was shown in the weight-bearing regions of the tibia [32]. In comparison, the mean cartilage thickness determined by MRI after 7 weeks of partial weight bearing in 20 patients with Weber B and C-type ankle fractures was reduced by −2.9% at the patella and −6.6% at the medial tibia [31]. Progressive joint cartilage thinning was also observed in 26 patients with complete, traumatic spinal cord injury, where the cartilage thickness of the patella, the medial, and the lateral tibia plateau was significantly and increasingly reduced during the first two years after injury [33]. Of note, in the patella, the mean cartilage thickness was found to be reduced by 23% at 24 months postinjury. The same authors reported longitudinal MRI results from spinal cord injury patients, where the mean change at 12 months was 9% in the patella, 11% in the medial tibia and in the medial femoral condyle, 13% in the lateral tibia, and 10% in the lateral femoral condyle [34].

In bed rest, COMP serum levels decreased by more than 20% and continued to decrease until 72 h into bed rest, while MMP-3 levels decreased within the first 24 h of bed rest [45,60]. Both COMP and MMP-3 levels recovered to baseline levels during a 6-day recovery period. MMP-1, MMP-9, and TNF-α levels were not affected. For future spacecraft and space stations, hypoxic conditions are under consideration, as a lower air pressure allows for much lighter construction. The addition of normobaric hypoxia to bed rest was shown to have no additional effect on the cartilage serum markers COMP and PIIANP [58].

In cell and tissue research, microgravity is often simulated in random positioning machines (RPM), where cultivated tissues showed intermediate characteristics compared with real microgravity on the International Space Station (ISS) and exposure to 1 G [61]. Cartilage tissue and cells have also been exposed to simulated weightlessness in rotating bioreactors, i.e., the clinostat and the rotating wall vessel (RWV) bioreactor as ground-based models for weightlessness and to produce spheroids, organoids, or tissues with and without scaffolds [62,63]. In rodent research, a common model used to simulate microgravity conditions that also shows intermediate characteristics is hindlimb unloading (HLU) [64,65].

## 4. Radiation Effects on Joint Cartilage

The radiation environment outside the Low Earth Orbit (LEO) consists of several types of ionizing radiation: the solar wind is a stream of charged particles released from the Sun and delivers a constant flux of radiation. In addition, solar particle events and coronary mass ejections may occur that release plasma clouds that mostly consist of electrons, protons, and alpha particles [66]. Moreover, galactic cosmic radiation that originates from outside our solar system is a particular threat to crew health, as it comprises high-energy protons and alpha particles, as well as high charge and energy (HZE) nuclei that move at relativistic speeds and energies and cause significant damage in the human body [66]. It is currently not possible to generate HZE particles for lab experiments on Earth, but the effects of some radiation types with lower energies on joint cartilage have been studied.

Exposure to ionizing radiation leads to chronic cartilage damage and impaired cartilage recovery, characterized by a degradation of cartilage, reduced proteoglycan synthesis, and impaired IGF-1 signaling [67,68]. Willey et al. [68] simulated exposure to the radiation of a solar-flare event in rats and compared a HLU group to a ground control group. The knee cartilage GAG content was lower in the HLU group, while both, HLU and total-body irradiation led to degenerative and pre-arthritic changes with incomplete recovery. In a different study with mice, radiation exposure groups of 0.1, 0.5, or 1.0 Gy during HLU were compared with a ground control group at 0 Gy (sham), and results indicated decreases in cartilage volume and thickness, increased serum COMP levels, and lower cartilage collagen content in all treatment groups compared with the control group [69]. Similar findings were reported for the long bones, but no study has to date been conducted for the subchondral bone. Lloyd et al. [70] studied the effect of 1 Gy of proton irradiation followed by mechanical unloading via HLU on bone in mice and showed that the radiation treatment alone led to approximately 20% loss of trabecular bone volume fraction in the tibia and femur. It thus seems likely that the subchondral bone is also affected.

## 5. Findings from Spaceflight

### 5.1. Human Research

Already in 1974, one of the astronauts who had spent 84 days on space station Skylab reported that ‘after return to one-g, the joints, especially the knees, felt sore after a little exercise’ [24]. So far, the only reported scientific human research on cartilage in real microgravity has been carried out in the ‘Cartilage’ experiment on the ISS (https://lsda.jsc.nasa.gov/Experiment/exper/1815, accessed on 5 June 2022). The results were reported in several abstracts as follows. Niehoff et al. [71] presented results of 10 astronauts who stayed on the ISS for five months or longer. Biomarkers for collagen type II degradation (cleavage of type II collagen, C2C) and synthesis (Procollagen II C-propeptide, CPII) were analyzed and showed high interindividual variation and increased serum levels at days 1, 7, and 30 post-landing, but the differences were statistically not significant. Preliminary results from 5 [71] and 6 astronauts [72] showed elevated serum COMP levels 7 and 30 days after return compared with pre-flight levels, but not on the first day after flight. In 12 crew members, Liphardt et al. [73] reported that the CTX-II concentration in the urine increased during the mission, reflecting increased cartilage degradation. Concentrations were highest during the first 4 months of the flight and then reached a plateau. Post-flight, urinary CTX-II levels were elevated compared with pre-flight and remained elevated for the first month after return to Earth. Within a year of landing, urinary CTX-II levels returned to normal in only 7 of the 12 crew members, indicating longer-term changes in cartilage physiology following spaceflight. Results from imaging studies on changes in cartilage thickness have not yet been reported after spaceflight, and it is thus not known which joints and joint cartilage regions are most affected by changes in cartilage thickness and composition.

Figure 1 shows the estimated trajectories of COMP serum and CTX-II urine concentrations during and after spaceflight, based on the findings summarized in this review.

### 5.2. Rodent Research

In rats flown for 18.5 days on Cosmos 1129, a Bion satellite mission in 1979, it was shown that the number of osteoblasts declined in the area immediately adjacent to the cartilage–metaphyseal junction, indicating changes in subchondral bone physiology [74]. In mice exposed to 30 days of microgravity in the Bion-M1 spacecraft in 1996, the articular cartilage of the hip and knee joints showed decreases in proteoglycan content post-flight compared with the control group [15]. Spaceflight and hindlimb unloading (HLU) were both shown to induce (osteo)arthritic responses in the knee cartilage of mice, as indicated by microCT, histology, proteomics, and further biochemical analyses [27]. Via microCT, cartilage thinning was shown to occur specifically at the weight-bearing tibia–femoral cartilage contact point after ~35 days in orbit, that was significantly more pronounced compared with ground and vivarium controls. The GAG content of the murine cartilage and menisci significantly decreased, while concentrations of catabolic enzymes (MMP-13) increased [27]. In the menisci, oxidative stress and catabolic molecular pathway responses were elevated, and after a 13-day Space Shuttle flight, meniscal degradation was observed [27]. During readaptation, recovery of cartilage volume and thickness occurred with exercise. Apart from cartilage and meniscal tissue, tibial epiphyseal plates of rats, that also contain cartilage cells, were studied after 12.5 days in space (Cosmos 1887), followed by 53.5 h recovery at 1 g. Flight animals had a larger proliferative zone with more cells compared to the controls [75].

### 5.3. Cell and Tissue Experiments

In vitro cultivation of cartilage cells on synthetic polymer scaffolds was conducted in a bioreactor on space station Mir [76]. Compared with the Earth group, the Mir-grown cartilage objects were more spherical, smaller, and mechanically inferior to those grown on Earth. Long-duration experiments on human chondroblasts were conducted with a 3-D porous biocarrier in the automated spacecraft Foton-M4 and showed that in microgravity, adhesive cells remained in the state of suspension [77]. On the ISS, scaffold-free neocartilage showed weaker extracellular matrix staining, higher collagen II/I expression ratios, lower aggrecan/versican gene expression profiles, and reduced cell density compared with 1 g ground control samples [61]. Cartilage cell density was significantly reduced in microgravity compared with normal gravity [61]. A reduced GAG concentration paired with alterations in collagen type 2 might be associated with an impaired resistance to mechanical stress [78]. In parabolic flights, that usually include 31 parabolas of 22 s of real microgravity, followed by a period of hypergravity each, human chondrocytes showed a significantly increased expression of anti-apoptotic genes after 31 parabolas, and increases in IL-6 in the supernatant during hypergravity [78]. The detailed effects of microgravity on stem cells have been reviewed elsewhere [79].

### 5.4. Deep-Space Missions

Until now, human spaceflight has usually taken place in the LEO, where spacecraft and space stations orbit the Earth [80]. Only during the Apollo missions, conducted by the United States of America in the late 1960s and early 1970s have humans left the LEO. In the LEO, the Earth’s magnetic field shields away most of the radiation coming from space, especially radiation types with higher energies [81]. Thus, radiation effects on the joint cartilage of spaceflight participants can be expected to be more pronounced on deep-space missions beyond the LEO, such as to the Moon and Mars, compared with findings obtained in crew staying on space stations that orbit Earth. As human spaceflight combines unloading and exposure to radiation, participants in deep-space missions are likely to experience a combination of the two effects. In five mice that left the LEO during the Apollo 17 mission in 1972 (the Apollo 17 Pocket Mouse Experiment, BIOCORE), no histological evidence of cosmic radiation effects was found in the vertebral column, femur, knee joint, tibia, and fibula [82]. However, no further detailed research results are currently available from beyond the LEO, as apart from the Pocket Mouse Experiment, no cartilage research has been conducted there to date.

## 6. Countermeasures

The ideal forces and types of stimulation to keep joint cartilage healthy during long-duration spaceflight are currently unknown and there is little literature available [83]. As it is also not known which joints and articular cartilage regions of human spaceflight participants are affected by unloading to what extent, it must be clarified for which joints countermeasures are required. As muscle and bone loss are most pronounced in the lower extremities, where the load difference to life on Earth is greatest, it seems likely that joint cartilage of the lower extremities will suffer more than the cartilage of the upper extremities [2,84]. In the spine, the greatest changes in intervertebral disc physiology during spaceflight were identified in the lumbar region where the load difference to Earth is greatest [85,86,87]. Regarding the type of loading, axial forces alone do not seem to be sufficient, as shear and more complex load scenarios are required to maintain a normal cartilage composition [56]. As crew time and payload are the most valuable resources in crewed spaceflight, it seems ideal to find countermeasures that address muscle, bone, and joint cartilage at the same time, e.g., by imitating locomotion and loading conditions on Earth. Currently, the 2 h daily exercise performed by the European astronauts on the ISS consists of resistive exercise using the Advanced Resistive Exercise Device, as well as treadmill running and cycling [88].

In the early years of spaceflight, no such in-flight exercise protocols or devices existed. Unfortunately, no data on changes in cartilage thickness or physiology exist from that time, neither from crew that did not exercise at all and may serve as a control group, nor from other exercise protocols prior to the current ones. From a scientific point of view, it would thus be desirable to study the effects of exposure to the spaceflight environment without exercise at least in a small group of spaceflight participants, and to use these control data to evaluate the efficacy of different exercise regimens and potentially other interventions on in-flight cartilage health. Data from bed rest studies may not be sufficient, as bed rest studies only reflect the effects of unloading but are unable to imitate the effects of space radiation on articular cartilage.

### 6.1. Human Centrifugation and Reactive Jumping

Artificial gravity by human centrifugation or in a rotating ring spaceship or space station is a potential countermeasure to mitigate the effects of weightlessness during long-term space missions, and its efficacy is currently under scientific evaluation [89,90]. In the spaceflight context, usually short arm human centrifuges with a diameter of around 7 m are discussed and studied that could fit inside a rocket or spaceship. Due to the short diameter, the participants’ heads are located close to the center of rotation, while their feet are the furthest away, resulting in a g-gradient within the body with the highest g levels at the feet. As an alternative to human centrifugation, the use of chronic artificial gravity in large rotating platforms to counteract microgravity pathologies is connected to several engineering challenges, but likely seems to be the most physiological solution [91,92]. In short arm human centrifuges, exposure to artificial gravity alone without further exercise is not sufficient to maintain the muscle mass, but studies testing the combined effects of artificial gravity and exercise are currently conducted [93,94,95]. Dreiner et al. [93] studied the effect of different types of impact loading under normal and artificial gravity conditions on serum COMP concentrations. Under Earth gravity, jumping and running resulted in a significantly higher increase in COMP serum level 30 min after impact loading than impact loading under artificial gravity. The authors also concluded that the number of impacts plays a crucial role for the cartilage metabolism, indicating that a higher number of impacts seems to be more beneficial.

### 6.2. Nutritional Countermeasures and Medications

Oral potassium bicarbonate and whey protein were tested as countermeasures in two 21-day bed rest studies of the European Space Agency but did not show effects on the cartilage biomarkers COMP, several MMPs, or TNF-α, that were determined before, during, and after bed rest [60]. Although Kartogenin has been suggested as a potential compound of an ‘exercise pill’ to promote chondrocyte differentiation, chondroprotection, and cartilage repair during long-duration space missions by improving chondrogenic differentiation of mesenchymal stem cells, further evidence and clinical data are needed as Kartogenin is not in clinical use [96]. Other potential countermeasures include fibroblast growth factor 18 (FGF-18) (Spifermin) [97], the selective cathepsin K inhibitor MIV-711 [98], and the Wnt pathway modulator SM04690 (Lorecivivint) [99] which all were shown to be structure modifying in early clinical trials. Sprifermin led to a dose-dependent reduction in cartilage thickness loss and cartilage thickness increase over 2 years. MIV-711 inhibits the endoprotease synthesized by osteoclasts and chondrocytes, which degrades the collagenous cartilage and bone matrix, and significantly reduces the progression of osteochondral OA changes after 26 weeks in vivo.

### 6.3. Vibration Training, Ultrasound, Shock Waves, and Magnetic Fields

In a 14-day bed rest study, vibration training with 2 × 5 min whole body vibration training/day increased the average cartilage thickness of the tibia by 21.9% and thus could be a potent countermeasure against reductions in articular cartilage thickness due to unloading [32]. Whole body vibration training (WBVT) in patients with knee OA was shown to strengthen the muscles around the knee and improve function and proprioception [100].

Low-intensity pulsed ultrasound (LIPUS) uses ultrasound waves to provide mechanical stimuli to inhibit inflammatory pathways associated with OA, and to activate molecular and cellular pathways in the cartilage to enhance chondrocyte proliferation, differentiation, and activity [101]. LIPUS has, however, not yet been studied in the spaceflight context, be it in real microgravity or in analogue studies. In case decreases in cartilage thickness and quality can be observed in spaceflight, LIPUS could be a potential countermeasure, and it should thus be studied further.

Extracorporeal shock wave therapy (ESWT) applies single pressure waves to promote biological healing processes through mechanotransduction. It was shown to improve cartilage and subchondral bone characteristics in OA [102]. Pulsed electromagnetic fields (PEMFs) were also shown to have positive effects on articular cartilage, subchondral bone, and synovia [102]. Just as LIPUS, both methods have not been used in spaceflight or analogue studies and might potentially be of interest as countermeasures during long-duration spaceflight in case of cartilage degeneration if conventional exercise does not suffice.

### 6.4. Radiation Protection

Since radiation damages articular cartilage, countermeasures against radiation during long-duration space missions could be of interest, at least for the most affected joints and as much as reasonably achievable. Protection from radiation can basically be achieved either by direct shielding, e.g., via polyethylene shielding, via storm shelters, especially shielded areas in the spacecraft where the crew hides when passing high-radiation zones, through electromagnetic shields, and via antioxidants to protect for oxidative injury caused by high-charge and high-energy particles [103,104,105]. While these concepts are known, it remains to be clarified which articular cartilage areas are affected and to what extent in order to make informed decisions about appropriate risk reduction.

### 6.5. The Role of Skeletal Muscle

Muscle and bone interact via physical forces and secreted osteokines and myokines [106]. Reduced muscular forces were shown to have a negative impact on the GAG content of joint cartilage, probably due to altered shear forces during unloading [56]. In addition to bed rest, hypoxia aggravates inactivity-related thigh muscle wasting in healthy active participants, as shown in the PlanHab bed rest study [107]. During 21 days of bed rest, thigh muscle size decreased by 6.9% in bed rest and by 9.7% when bed rest was combined with normobaric hypoxia. Muscle volume decreases during bed rest, immobilization, and spaceflight. In a 60-days bed rest study, the 8 female participants in the bed rest-only group lost 21% of their quadriceps femoris muscle mass [108], and 9 men undergoing 90 days of bed rest lost 18% of their knee extensor and 29% of their plantar flexor muscle volume [109]. LeBlanc et al. [110] measured losses in muscle volume in several space shuttle missions and found decreases by 3–10% in 17-day space shuttle missions, and 5–17% in the longer flights of the Shuttle/MIR program. In astronauts returning from space, changes in coordination were observed, such as impaired head-trunk coordination, increased sway, and risk of tripping [111,112,113]. Such disturbances in neuromuscular interaction might affect joint alignment and thus lead to cartilage damage. Healthy muscle tone and power are therefore desirable to keep the cartilage intact upon re-loading.

## 7. Discussion

This review summarized the current literature available on joint cartilage alterations in long-duration spaceflight as condensed in Figure 2. Evidence from spaceflight participants is limited to serum biomarker data in only a few astronauts. Findings from analogue model research and simulated microgravity, such as bed rest studies, as well as data from animal and cell research in real microgravity indicate that unloading and radiation exposure are associated with joint degeneration in terms of cartilage thinning and changes in cartilage composition. It is currently unknown how much the individual cartilage regions in the different joints of the human body will be affected on long-term missions beyond the Low Earth Orbit. The use of countermeasures that put either load and shear, or other stimuli on the joints, shield them from radiation, support cartilage physiology, or remove oxidative stress possibly help avoid OA in later life following long-duration space missions. There is, however, a high demand for research on the efficacy of such countermeasures to judge their suitability for their implementation in long-duration spaceflights.

When planning this research, analogue studies and simulated microgravity will play an important role, but their limitations should be considered. The main limitation is the lack of exposure to space radiation when performing human studies on Earth, despite the high relevance of radiation for cartilage health as described in Section 4. First results on joint cartilage biomarker recovery seem to indicate that after bed rest, COMP serum levels seemed to recover to baseline faster [60] than after real spaceflight [71,72]. However, due to the lack of larger data sets, these first findings need to be seen with a lot of caution, and further studies are required that study differences in findings between simulated microgravity on Earth and the real spaceflight environment. Since several types of ionizing radiation make up the radiation environment outside the LEO, at least some radiation types should be added to the immobilization or microgravity models when planning analogue studies on Earth to study articular cartilage for long-duration space missions, as previously done by some groups [68,69,70]. The radiation applied in such studies could consist of charged particles, electrons, protons, and alpha particles. Ideally, articular cartilage research will be conducted in humans on missions to the Moon, e.g., making use of the planned space station Gateway and missions to the lunar surface.

Gravity levels are lower on the Moon and on Mars compared with Earth, which makes skipping and hopping the preferred types of bipedal locomotion for humans, as they are more efficient and less fatiguing than walking or running [9,114,115]. Skipping and hopping are associated with altered forces acting on the human musculoskeletal system compared with on Earth. Differences were identified, i.e., for mechanical and muscle work, ground reaction forces, and joint angles [116]. Impact forces that act on the human joints are expected to be lower in partial gravity. It might therefore be necessary to implement countermeasures for musculoskeletal health, i.e., exercise countermeasures with a high speed of locomotion, such as sprinting, despite the existence of at least some gravity [117]. As the Moon and Mars neither have magnetic fields nor a sufficiently dense atmosphere like the Earth, radiation exposure of galactic cosmic rays and solar energetic particles will be as high on the surface as during transit [25,118]. Radiation effects on human joint cartilage can therefore be expected to be of great magnitude, and due to a lack of data, attention should be given to joint cartilage alterations in the first humans who will leave the Earth’s magnetosphere since the Apollo program.

## 8. Conclusions

Knowledge on joint cartilage alterations in human spaceflight is very limited, but based on evidence from cell, animal, and human research, the exposure to microgravity combined with radiation is likely to lead to joint cartilage thinning and degeneration, and consequently to OA after long-term space missions. In view of these expected issues, studies on joint cartilage alterations in human spaceflight participants, as well as in analogue studies, and research on countermeasures are urgently recommended, especially with regard to the planning of long-term missions to the Moon and Mars.

## Figures and Tables

**Figure 1 biomedicines-10-01356-f001:**
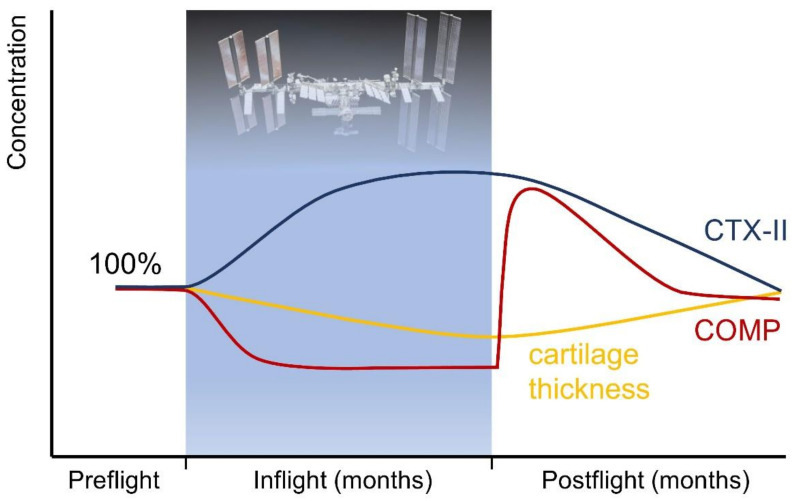
Estimated trajectories of human COMP serum and CTX-II urine concentrations, as well as of cartilage thickness during and after spaceflight based on the findings summarized in this review. COMP and CTX-II might be candidates to monitor the efficacy of in-flight countermeasures to prevent cartilage degradation.

**Figure 2 biomedicines-10-01356-f002:**
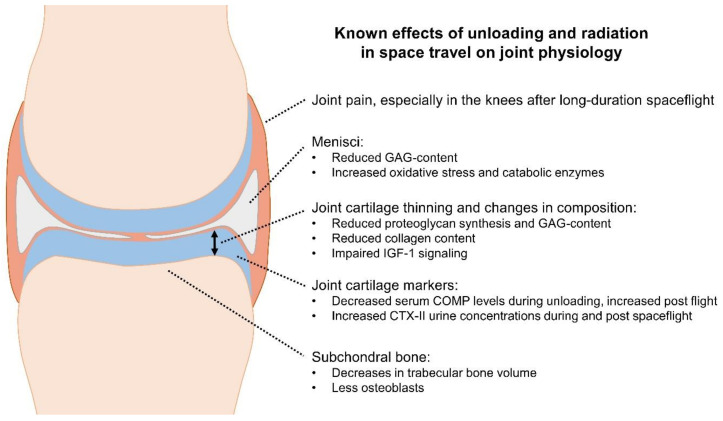
Illustration of the main known effects of unloading and radiation on joint physiology.

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
