# Peer review of "Joint Cartilage in Long-Duration Spaceflight"

_biomedicines, 2022, doi:10.3390/biomedicines10061356_

Round 1
Reviewer 1 Report
The review manuscript by Ganse et al: “Joint cartilage in long-duration spaceflight” is an interesting paper providing a good overview regarding the knowledge, or maybe more the lack of knowledge for humans, regarding the impact of spaceflight on articular cartilage.
Only a few comments / suggestions for this manuscript:
Line 45-48: It is stated that “Clinical musculoskeletal problems during spaceflight include….. bends and dysbaric osteonecrosis…” This statement seems quite bold to me. Has any bends or osteonecrosis been reported for cosmonauts/astronauts? Is this a clinical problem or maybe more of possible operational risk?
Figure 1:
Could authors complement / add some more detailed time indications on the x-axes, at least more detailed than just pre-post and in-flight.
Also, could other estimated values related to cartilage metabolism / catabolism be included ?
Line 205: 5.2 Rodent data:
Translational data: Attention is given to human model for near weightlessness simulation (bed rest/dry immersion) but, besides effects of radiation and HLU, no mention is made for this animal model regarding only the biomechanical side of possible cartilage reduction/damage. Data from such studies might / are of interest also for humans. So, consider including papers like doi: 10.1007/s00776-005-0931-7, doi: 10.1177/19476035211063857 or doi: 10.1371/journal.pone.0254383.eCollection 2021
Line 222: 5.3. Cell and tissue experiments
In vitro data:
Group from Duke / Montufar-Solis : e.g. 10.1096/fasebj.4.1.2295377 and group from Veldhijzen from Shuttle STS-42.
Line 255: 6 Countermeasures: / 6.1. Human centrifugation and reactive jumping
The authors mention the possible application of a short-arm centrifugation with additional exercise modalities to mitigate microgravity-related effects. One should also consider the use of chronic artificial gravity to counteract microgravity pathologies. Such large rotating platforms have always been framed by, currently out-of-date, engineering and/or budget arguments. However, current studies indicate that a rotating spacecraft is very well feasible and would very likely solve the issue of lack of mechanical load to the body (see e.g. DOI: 10.2514/6.2020-4111, DOI: 10.1016/j.actaastro.2015.11.030, and/or DOI: 10.3389/fphys.2020.00470)
Also this should be addressed as possible countermeasure.
Line: 379: Please check grammar
Line 383: “..lack 383 of current evidence,..”: Is this lack of evidence or lack of data?
General remarks:
The SI unit for gravity is ‘g’ (lower case g). Capital “G” is incorrect since is stands for gravitational constant and not for gravity
Author Response
Reviewer 1:
The review manuscript by Ganse et al: “Joint cartilage in long-duration spaceflight” is an interesting paper providing a good overview regarding the knowledge, or maybe more the lack of knowledge for humans, regarding the impact of spaceflight on articular cartilage.
Only a few comments / suggestions for this manuscript:
Line 45-48: It is stated that “Clinical musculoskeletal problems during spaceflight include….. bends and dysbaric osteonecrosis…” This statement seems quite bold to me. Has any bends or osteonecrosis been reported for cosmonauts/astronauts? Is this a clinical problem or maybe more of possible operational risk?
Response: Indeed, bends are a well-known clinical problem in spaceflight and have occurred during numerous space missions, e.g., during Gemini 10 and Apollo 11 [Johnston RS, Dietlein LF. Biomedical Results from Skylab. Scientific and Technical Information Office, National Aeronautics and Space Administration, SP-377, Washington, 1977.]. Due to countermeasures, such as pre-breathing before EVAs, the incidence has been lower in recent years, but problems would immediately show up if these interventions we not performed anymore, or if a micro-meteoroite hit a spacecraft or space station. We have added this reference to the paragraph indicated by the reviewer. We have also changed ‘occurs’ to ‘occurred’, as decompression events have been less frequent in recent years.
Figure 1:
Could authors complement / add some more detailed time indications on the x-axes, at least more detailed than just pre-post and in-flight.
Also, could other estimated values related to cartilage metabolism / catabolism be included ?
Response: We have added the time information to the x-axis. Evidence from human spaceflight is only available for COMP and CTX-II, and we do not want to be too speculative about the other parameters, as differences in the COMP trajectory were observed between bed rest studies and real microgravity. But we have added a speculative trajectory of cartilage thickness.
Line 205: 5.2 Rodent data:
Translational data: Attention is given to human model for near weightlessness simulation (bed rest/dry immersion) but, besides effects of radiation and HLU, no mention is made for this animal model regarding only the biomechanical side of possible cartilage reduction/damage. Data from such studies might / are of interest also for humans. So, consider including papers like doi: 10.1007/s00776-005-0931-7, doi: 10.1177/19476035211063857 or doi: 10.1371/journal.pone.0254383.eCollection 2021
Response: We have added a sentence on the HLU model to the end of paragraph 3 that cites two of the three papers suggested by the reviewer.
Line 222: 5.3. Cell and tissue experiments
In vitro data:
Group from Duke / Montufar-Solis : e.g. 10.1096/fasebj.4.1.2295377 and group from Veldhijzen from Shuttle STS-42.
Response: We have added this information to section 5.2.
Line 255: 6 Countermeasures: / 6.1. Human centrifugation and reactive jumping
The authors mention the possible application of a short-arm centrifugation with additional exercise modalities to mitigate microgravity-related effects. One should also consider the use of chronic artificial gravity to counteract microgravity pathologies. Such large rotating platforms have always been framed by, currently out-of-date, engineering and/or budget arguments. However, current studies indicate that a rotating spacecraft is very well feasible and would very likely solve the issue of lack of mechanical load to the body (see e.g. DOI: 10.2514/6.2020-4111, DOI: 10.1016/j.actaastro.2015.11.030, and/or DOI: 10.3389/fphys.2020.00470)
Also this should be addressed as possible countermeasure.
Response: We are thankful for the reviewer’s suggestion and have added artificial gravity through rotating spacecraft or space stations to section 6.1, including the first two of the references.
Line: 379: Please check grammar
Response: Fixed
Line 383: “..lack 383 of current evidence,..”: Is this lack of evidence or lack of data?
Response: We have changed it to ‘lack of data’.
General remarks:
The SI unit for gravity is ‘g’ (lower case g). Capital “G” is incorrect since is stands for gravitational constant and not for gravity
Response: We have changed it throughout the manuscript.
Reviewer 2 Report
The manuscript is extremely well organized and well written.
Comments for minor revision:
1. On page 4, section 4: define LEO. (There is a definition of LEO on page 6, line 240, so this should be moved up.) Also in this sentence, the "(linear energy transfer, LET)" is not needed, and can be removed from this sentence.
2. Page 6, line 252: Remove the word "more", since it is redundant.
3. Page 6, section 6: Please comment further on the duration of exercise currently being required for in flight astronauts, and whether there are any comparisons with previous flights that did not have exercise requirements. Were there any improvements in cartilage with the required exercises?
4. Sections 7.1 under the discussion seem out of place, since this section are still portions of text that review the literature and published data. It seems like this might be a countermeasure topic? Section 7.2 does seem to be a discussion section, but does not necessarily need its own heading.
Author Response
Reviewer 2:
The manuscript is extremely well organized and well written.
Comments for minor revision:
- On page 4, section 4: define LEO. (There is a definition of LEO on page 6, line 240, so this should be moved up.) Also in this sentence, the "(linear energy transfer, LET)" is not needed, and can be removed from this sentence.
Response: Thanks for pointing that out! We have done so.
- Page 6, line 252: Remove the word "more", since it is redundant.
Response: Done.
- Page 6, section 6: Please comment further on the duration of exercise currently being required for in flight astronauts, and whether there are any comparisons with previous flights that did not have exercise requirements. Were there any improvements in cartilage with the required exercises?
Response: We are thankful for this valuable addition and have added a paragraph on these issues to the end of the first paragraph of section 6.
- Sections 7.1 under the discussion seem out of place, since this section are still portions of text that review the literature and published data. It seems like this might be a countermeasure topic? Section 7.2 does seem to be a discussion section, but does not necessarily need its own heading.
Response: As requested by the reviewer, section 7.1 was moved to the end of section 6. In addition, the heading of section 7.2 was removed.
Reviewer 3 Report
The submitted manuscript is a very interesting example of the article taking into consideration cartilage physiology under space conditions. Introduction part including spaceflights is very interesting - I would only add few sentences about the number of articles about cartilage physiology in space to make sure that it is very narrow field of science.
General effects of real microgravity are presented well, however, in the Reviewer's opinion the article is missing about the effects of simulated microgravity. Please add some paragraphs about that and compare the findings with real microgravity studies.
The Reviewer really appreciated the parts about the biomarkers of joint cartilage unloading. Maybe the Authors could prepare some kind of figure to present such information in a little more interesting way?
Summarizing, I recommend this manuscript for publication when the Authors develop the abovementioned issues in the article.
Author Response
Reviewer 3:
The submitted manuscript is a very interesting example of the article taking into consideration cartilage physiology under space conditions. Introduction part including spaceflights is very interesting - I would only add few sentences about the number of articles about cartilage physiology in space to make sure that it is very narrow field of science.
Response: We have added this information as suggested and added the following sentences to the introduction: ‘Only very few articles are available that deal with cartilage physiology in space. A Pubmed search for the terms ‘spaceflight joint cartilage’ only revealed 19 hits on June 5, 2022.’
General effects of real microgravity are presented well, however, in the Reviewer's opinion the article is missing about the effects of simulated microgravity. Please add some paragraphs about that and compare the findings with real microgravity studies.
Response: We are thankful for the input and have added a longer paragraph on this issue to the discussion section, as well as information on the differences in findings and on future research, as suggested by the reviewer. We have also added ‘as well as in analogue studies’ to the conclusions.
The Reviewer really appreciated the parts about the biomarkers of joint cartilage unloading. Maybe the Authors could prepare some kind of figure to present such information in a little more interesting way?
Response: There are actually many papers around that deal with these biomarkers. We have now added the cartilage thickness to Figure 1 and improved Figure 2 a bit, too.
Summarizing, I recommend this manuscript for publication when the Authors develop the abovementioned issues in the article.